# Depression, suicidal ideation and suicide risk in German veterinary medical students compared to the German general population

**Nadine Schunter** [1]*, **Heide Glaesmer**[2], **Luise Lucht**[2], **Mahtab Bahramsoltani**[1]

**1** Institute of Veterinary Anatomy, Freie Universität Berlin, Berlin, Germany, **2** Department of Medical Psychology and Medical Sociology, University of Leipzig, Leipzig, Germany

\* nadine.schunter@fu-berlin.de

## Abstract

### Background

Various studies from different countries indicated that veterinarians have a significantly increased risk of depression, suicidal ideation and of death by suicide. For German veterinarians a recent study has demonstrated a three times higher rate for depression, two times higher rate for suicidal ideation and a five times higher suicide risk compared to the German general population. For veterinary students, recent studies in the US and UK indicated higher mental distress. To date, empirical studies on depression, suicidal ideation and suicide risk among veterinary students in Germany were lacking so far. This study investigates depression, suicidal ideation and suicide risk of veterinary students in Germany.

### Methods

913 German veterinary students (14.3% response rate, 90.7% women, mean age 23.6 years) between 18 and 46 years were included and compared with representative German general population samples from 2007 (N = 1097, 55.4% women, mean age 33.9) and 2015 (N = 1033, 56.1% women, mean age 32.8) of the same age range using the depression module of the Patient Health Questionnaire (PHQ-9) and Suicide Behaviors questionnaire-Revised (SBQ-R). The general population samples were collected with the assistance of a demographic consulting company.

### Results

The prevalence of depression among German veterinary students was 45.9% (compared to 3.2% in the general population), suicidal ideation was 19.9% (compared to 4.5% in the general population) and suicide risk was 24.0% (compared to 6.6% in the general population).

### Conclusion

In this study, German veterinary students have a 22.1 times higher risk to be screened positive for depression, a 4 times higher risk for reporting current suicidal ideation and they are

**Funding:** The authors received no specific funding for this work.

**Competing interests:** The authors have declared that no competing interests exist.

4.2 times more likely to have an increased suicide risk compared with the general population in Germany of the same age range.

## Introduction

Various studies from several countries have reported an up to four times higher risk to die by suicide in veterinarians compared to the general population [1–5]. In 2016 Schwerdtfeger et al. conducted the first study on depression, suicidal ideation and suicide risk among veterinarians in Germany [6]. The results indicated a more than six times higher risk of suicide, two times higher rates of suicidal ideation and three times higher rates of depression in veterinarians compared to the German general population. Unfortunately, less is known about specific causes for this remarkably increased risk. Generally, mental disorders (e.g. depression) are described as common risk factors of suicidal ideation and behavior [7, 8]. Among veterinarians specific risk factors have been addressed in recent studies, such as personality traits, professional and social isolation [1, 9], job-related stressors like heavy workload, long working hours, nightshifts and weekend services, poor work-life balance, difficult or challenging interactions with clients, clients' expectations [1, 5, 10, 11], access to and knowledge about lethal medications [11–14] and unique work experiences such as euthanasia [11, 15, 16].

Since the high rates of depression, suicidal ideation and suicide risk in veterinarians were found, recent studies are investigating the mental distress of veterinary students enrolled in the veterinary program [17–27]. Hafen et al. [23] found evidence that veterinary students experience higher rates of depression than students in other professional programs. For first-year veterinary students in the US Hafen et al. showed in both a cross-sectional investigation [23] and a longitudinal investigation [25] that one-third of the surveyed students had symptoms of depression. Studies in US veterinary students in all years of study indicated that one-third [20, 27] up to approximately two-third [17] of the participants suffer from depressive symptoms. Additionally, the results of recent studies demonstrate that the prevalence of depression among US veterinary students is higher than those in the general population of the US [22, 23].

For suicidal ideation and suicide attempts, Cardwell et al. stated that UK veterinary undergraduates show significantly higher rates of suicidal ideation, but significantly lower rates of suicide attempts compared to the general population [26]. Furthermore, for veterinary students in the US Karaffa and Hancock found that almost one-third of the surveyed students reported having had seriously thought about suicide [27].

There are possible reasons discussed in the literature for the elevated risk of depression, suicidal ideation and suicide risk in veterinary students. According to Brscic et al., the risk factors stated for veterinarians, like performing animal euthanasia and access to lethal drugs, as well as managerial aspects of the job and difficult interactions with clients are primally specific for veterinary practitioners and could not be readily applied to veterinary students [11]. However, the heavy workload and continuous examination load, unclear professor expectations, chronic sleep deprivation, lack of time for social and recreational activities, unsatisfactory family and personal relationships, difficulties fitting in with peers, feeling being behind in their studies, concerns about one's own academic performance and a fear of failing were discussed as common stressors [19, 22, 23, 25]. As a result of the feeling from not fitting in with peers [25], being behind in studies [19] and having a fear of failing [20], many students begin to sacrifice private areas of their lives in order to have more time to study [20]. Chronicity of these feelings during the whole training can lead to a decreased study-life balance and social isolation as well

as stress, anxiety and depression [18–20]. Consequently, a diminished feeling of belonging to the student community and professional belonging can emerge from it [21].

Cardwell and Lewis found in their study that belonging to both the student and professional community played an important role among the students they interviewed [21]. They referred to Joiner's Interpersonal Psychological Theory of Suicide (IPTS), in which thwarted belongingness is a core component for the emergence of suicidal ideation [28].

To date, there is a large amount of studies from several countries [17–27], but no empirical evidence on depression, suicidal ideation and suicide risk among veterinary students in Germany. The aim of this study was to investigate rates of depression, suicidal ideation and suicidal behavior in German veterinary students in comparison to the German general population.

## Method

### Sample of German veterinary students

To recruit as many German veterinary students as possible, the invitation to participate in the study was distributed through the Dean's Offices and Study Offices of the five German veterinary schools between November and December 2018. The Study Offices sent the link to the questionnaire via e-mail to all the students enrolled. In addition, the veterinary student council initiatives of the respective veterinary schools, that are composed of the elected representatives of the corresponding veterinary students enrolled and are representing their interests, were contacted and asked to distribute the invitation to participate in the survey among the students. In March 2019 the students were reminded of the survey via e-mail including a link to the questionnaire and invited again to answer the questionnaire.

Inclusion criteria were sufficient knowledge of the German language and enrolment as veterinary medical student in one of the five German veterinary schools. Potential participants were invited to answer an anonymous online questionnaire via Unizensus (Blubbsoft GmbH) between November 2018 and April 2019. Through the email cover letter and the information provided by the Dean's Offices, Study Offices and veterinary student council initiatives, the invited students were informed about the study and its objectives. The information text preceding the questionnaire informed the study participants about the aim of the study, possible advantages and risks of participation as well as data protection aspects. This was followed by the consent form, in which the study participants clearly agreed or refused to the study. This consent form was a mandatory question in multiple-choice format to be answered. If the study participant agreed to the consent form, the questionnaire started. If the study participant refused the consent form, the survey was terminated immediately. The Ethics Commission at the Medical Faculty of the University of Leipzig (EC No. 139/18-ek) approved the study protocol.

A total of 922 students participated in the survey. Participants with more than 25% missing data per scale or unreliable information about depressive symptoms, suicidal ideation and suicidality (n = 9) were excluded from the analysis. Therefore, 913 German veterinary students (90.7% female, n = 828, mean age 23.6 years, *SD* = 3.96) between the age of 18 years and 46 years were integrated into the analyses. Thirty-seven participants did not provide any information about their age. One participant indicated diverse gender. This person was accordingly not assigned to the male/female group analyses, but was included in the overall analyses. Based on the 2018 statistics report of the Federal Chamber of Veterinarian Medicine in the winter term 2018/2019 6,367 veterinary students were enrolled at the five German veterinary schools that took part in the survey [29], this corresponds to a response rate of 14.3%.

For comparing our findings with the German general population, data from two representative general population samples collected in 2007 (Hauffa et al. [30]) and 2015 (Glaesmer et al. [31]) were used as reference data. Using the representative general population samples aim to compare the prevalence of depression, suicidal ideation and suicide risk within the German general population with the results of the sample of German veterinary students of the same age range.

## General population samples

Two representative samples of the German general population were collected with the assistance of a demographic consulting company (USUMA, Berlin, Germany) in 2007 and in 2015. For both samples, the German territory was separated into 258 sample areas representing the different regions of the country. Households of the respective areas and members of these households fulfilling the inclusion criteria (age at or above 14 years and able to read and understand the German language) were selected randomly. Both samples are representative in terms of age, gender and education.

In 2007, a first attempt was made for 4,205 addresses, of which 4,055 turned out to be valid. In 2015, a first attempt was made for 4,902 addresses, of which 4,844 turned out to be valid. If unsuccessful at first, a maximum of three attempts was made to contact the selected person. All subjects were visited by a study assistant, informed about the study and handed self-rating questionnaires. The assistant waited until participants answered all questions and offered help, if persons did not understand items.

For general population sample 1, data collection took place in May and June 2007. A total of 2,510 people between the ages of 14 and 93 (54.5% female) agreed to participate and completed self-rating questionnaire (participation rate: 61.9% of valid addresses). A total of 1,097 participants aged 18–46 years (55.4% female; mean age 33.9 years) were used as a general population comparison group. The PHQ-9 was used in this study. For more details see Hauffa et al. [30].

For general population sample 2, data collection took place between March and May 2015. A total of 2,513 people between the ages of 14 and 94 (55.5% female) agreed to participate and completed the self-rating questionnaire (participation rate: 61.9 per cent of valid addresses). 1,033 participants aged 18 to 46 years (56.1% female; mean age 32.8 years) were used as a general population comparison group. The study was reviewed and approved by the ethics committee of the German Psychological Association. The PHQ-9 and the SBQ-R were used in this study. For more details see Glaesmer et al. [31].

## Instruments

The online questionnaire used with the veterinary student population for this research comprises items about (1) demographics (e.g. gender, age, gainful employment while studying, children, semesters), (2) study conditions (e.g. time of study choice, knowledge of study requirements, social climate between students and lecturers, support and supervision by teachers), (3) several standardized instruments to assess depression [32, 33] and suicidality [34, 35] and (4) standardized questionnaires to assess suicide specific constructs from the Interpersonal Psychological Theory of Suicide [36–40], and Effort-Reward-Imbalance [41], as well as personality traits [42], general life satisfaction [43], learning and achievement motivation and recognition during studies [44, 45]. The two instruments (PHQ-9 and SBQ-R) analyzed for this study are described in more detail below.

## Depression

The Depression module of the Patient Health Questionnaire (PHQ-9) is a self-report measure for the assessment of depressive symptoms and suicidal ideation [32, 33]. The PHQ-9 assesses nine different depressive symptoms according to the Diagnostic and Statistical Manual of Mental Disorders, 4th Edition (DSM-IV) on a 5-point-Likert-Scale ranging from 0 ('not at all') to 3 ('nearly every day'). The total score is calculated from the nine individual items and suggests the following levels of depression: 0–4 = none to minimal, 5–9 = mild, 10–14 = moderate, 15–19 = moderately severe and 20–27 = severe depression. The total scores range from 0 to 27. Scores of 10 and above indicate at least relevant depressive symptoms.

## Suicidal ideation

Suicidal ideation was defined as being bothered by thoughts, that oneself would be better off dead, or of hurting oneself in some way, over the past two weeks. Therefore, item 9 of the PHQ-9 ('Thoughts that you would be better off dead, or of hurting yourself in some way') was applied to assess suicidal ideation. If item 9 was answered with scores from 1 (several days) to 3 (nearly every day), participants were classified as having current suicidal ideation.

## Suicide risk

To reliably and validly assess different aspects of suicidality, the Suicide Behaviors Questionnaire-Revised (SBQ-R) was applied [34, 35]. The SBQ-R is recommended for population-based studies [46]. Additionally, it is suggested for use in adolescent and adult psychiatric inpatient as well as in nonclinical adolescent and adult participant [34]. The SBQ-R consists of four items. Each item describes a specific aspect of suicidal experience and behavior: life-time suicidal ideation, plans and attempts; frequency of suicidal ideation over the last 12 months; threat of suicidal attempt and likelihood of suicidal behavior in the future [28]. Only one answer option should be selected per item. The total score is calculated, which range between 3 and 18. The higher the total score, the greater the probability of suicidal behavior. To identify participants with increased suicide risk, a total score of 7 and above is used [34].

## Statistical analyses

Statistical analyses were conducted with IBM SPSS 24.0 for Windows.

For comparing prevalence rates of depression, suicidal ideation, and suicide risk in veterinary students and the general German population, binary logistic regression analyses (controlled for age and gender) were applied. We employed a tolerance criterion of 25% missing values per scale. We excluded a participant's data for the respective scale, if they responded to less than 75% of the items. The remaining missing data per scale were then substituted with the individual mean score (person mean imputation).

# Results

## Study sample of German veterinary students

Our study sample comprised 913 German veterinary students aged 18 to 46 years (90.7% female, n = 828, mean age 23.6 years) and represents 14.3%, (85.4% female, n = 5,440) of the entire population of German veterinary students (N = 6,367) based on the 2018 statistics report of the Federal Chamber of Veterinarian Medicine [29]. Compared to the percentage of females within the entire population of German veterinary students, the proportion of female veterinary students in our sample is approximately 5% higher (85.4% female of the entire population of German veterinary students vs. 90.7% female in this study). The higher proportion of

women in our study might be due to the fact, that women tend to report more mental distress than men [47–49].

## Depression, suicidal ideation and suicide risk in German veterinary students compared to the German general population

Table 1 presents the prevalence of depression, suicidal ideation and suicide risk in the sample of German veterinary students and in a representative sample of the German general population of the same age range.

Symptoms of depression were assessed with the PHQ-9 using an established cut-off-score of $\geq$10 (moderate to severe). According to this cut-of-score, 45.9% (n = 416) of the veterinary students were screened positive for depression. Of these, 22.6% (n = 205) showed moderate symptoms of depression and 23.3% (n = 211) indicated moderately severe to severe symptoms

**Table 1. Depression, suicidal ideation and suicide risk in veterinary students and the German general population.**

| | | Veterinary Students | | | | | | General Population[1,2] | | | | | | Statistical tests |
|---|---|---|---|---|---|---|---|---|---|---|---|---|---|---|
| | | Female | | Male | | Total | | Female | | Male | | Total | | Vetstud vs. general population[3] |
| | | n = 828 | | n = 84 | | N = 913[4] | | n = 605 | | n = 489 | | N = 1094 | | |
| **Level of Depression** | | | | | | | | | | | | | | |
| **PHQ-9** | | m | sd | m | sd | m | sd | m | sd | m | sd | m | sd | |
| | | 10.04 | 5.99 | 7.54 | 5.76 | 9.81 | 6.01 | 2.37 | 3.34 | 1.66 | 2.95 | 2.05 | 3.19 | |
| Severity | Minimal | 164 | 19.9 | 32 | 38.6 | 196 | 21.6 | 496 | 82.0 | 433 | 88.5 | 929 | 84.9 | |
| | Mild | 273 | 33.1 | 24 | 28.9 | 297 | 32.7 | 88 | 14.5 | 42 | 8.6 | 130 | 11.9 | |
| | Moderate | 188 | 22.8 | 16 | 19.3 | 205 | 22.6 | 14 | 2.3 | 8 | 1.6 | 22 | 2.0 | |
| Moderately severe to severe | | 200 | 24.2 | 11 | 13.2 | 211 | 23.3 | 7 | 1.2 | 6 | 1.2 | 13 | 1.2 | |
| Total score moderate to severe > = 10 | | 388 | 47.0 | 27 | 32.5 | 416 | 45.9 | 21 | 3.5 | 14 | 2.8 | 35 | 3.2 | OR = 22.076 |
| | | | | | | | | | | | | | | (CI = 13.873–35.131)*** |
| **Suicidal Ideation** | | | | | | | | | | | | | | |
| **PHQ-9 / Item 9** | | Female | | Male | | Total | | Female | | Male | | Total | | |
| Not at all | | 655 | 79.4 | 73 | 86.9 | 729 | 80.1 | 572 | 94.5 | 473 | 96.7 | 1045 | 95.5 | |
| Several days | | 121 | 14.7 | 6 | 7.1 | 127 | 14.0 | 27 | 4.5 | 14 | 2.9 | 41 | 3.7 | |
| More than half the days | | 35 | 4.2 | 2 | 2.4 | 37 | 4.1 | 3 | 0.5 | 1 | 0.2 | 4 | 0.4 | |
| Nearly every day | | 14 | 1.7 | 3 | 3.6 | 17 | 1.9 | 3 | 0.5 | 1 | 0.2 | 4 | 0.4 | |
| Sum > 0 | | 170 | 20.6 | 11 | 13.1 | 181 | 19.9 | 33 | 5.5 | 16 | 3.3 | 49 | 4.5 | OR = 3.960 |
| | | | | | | | | | | | | | | (CI = 2.542–6.168)*** |
| **Suicide Risk** | | | | | | | | | | | | | | |
| **SBQ-R[5]** | | Female | | Male | | Total | | Female | | Male | | Total | | |
| | | | | | | | | n = 578 | | n = 453 | | N = 1031 | | |
| >7 (Suicide Risk) | | 191 | 24.4 | 17 | 21.3 | 208 | 24.0 | 42 | 7.3 | 26 | 5.7 | 68 | 6.6 | OR = 4.166 |
| | | | | | | | | | | | | | | (CI = 2.829–6.135)*** |
| = <7 (none/minimal Suicide Risk) | | 593 | 75.6 | 63 | 78.8 | 657 | 76.0 | 536 | 92.7 | 427 | 94.3 | 963 | 93.4[4] | |

1 Study participants over the age of 46 years were excluded

2 Representative samples from the German general population from 2007 (depression, suicidal ideation) and 2015 (SBQ-R, suicide risk)

3 Binary logistic regression including groups (general population as reference category vs. veterinary students) controlled for age and gender with depression/suicidal ideation/suicide risk as dependent variables, n = 37 veterinary students were excluded from the analysis due to missing data concerning their age

4 One participant with diverse gender was included in the overall analyses ("total")

5 Suicide Behaviors Questionnaire, maximum score of 18 possible

of depression. In comparison, only 3.2% (n = 35) of the German general population of the same age range were screened positive for depression, with 2.0% (n = 22) showing moderate and 1.2% (n = 13) showing moderately severe to severe symptoms of depression. To test the differences between our study sample and the general population, we conducted a binary logistic regression analysis. After controlling for age and gender, German veterinary students have a 22.1 times higher risk to be screened positive for depression compared with the German general population (OR = 22.08; 95% CI = 13.87–35.13).

Suicidal ideation was assessed using Item 9 of the PHQ-9. In the sample of veterinary students 19.9% (n = 181) answered item 9 with scores from 1 (several days) to 3 (nearly every day) and are therefore classified as having suicidal ideation in the past two weeks. Among them a majority of 14.0% (n = 127) reported to have had such feelings on several days during the last two weeks, 4.1% (n = 37) on more than half of the days and 1.9 (n = 17) nearly every day during the last two weeks. In the German general population in only 4.5% (n = 49) suicidal ideation was found, with 3.7% (n = 41) indicated this on several days, only 0.4% (n = 4) on more than half of the days and only 0.4% (n = 4) showed suicidal ideation on nearly every day during the last two weeks. To test the differences between our study sample and the general population, we conducted a binary logistic regression analysis. After controlling for age and gender, German veterinary students have an almost 4 times higher risk for having current suicidal ideation compared with the German general population (OR = 3.96; 95% CI = 2.54–6.17).

Suicide risk was assessed using the SBQ-R using cut-off-score of >7 (increased suicide risk). Among German veterinary students, 24.0% (n = 208) were classified as having an increased suicide risk. In comparison, only 6.6% (n = 68) the German general population were classified as having an increased suicide risk. To test the differences between the study sample and the general population, we conducted a binary logistic regression analysis. After controlling for age and gender, German veterinary students appeared to be 4.2 times more likely to have an increased suicide risk than the German general population (OR = 4.17; 95% CI = 2.83–6.14).

## Discussion

Recent studies from several countries provide evidence that veterinarians have a significantly increased risk of suicidal ideation and behavior compared to the general population [1–5]. The first study on depression, suicidal ideation and suicide risk among veterinarians in Germany was conducted by Schwerdtfeger et al. in 2016. The results confirm the findings of previous studies from several countries. According to Schwerdtfeger et al., German veterinarians have a remarkably increased risk of depression, suicidal ideation and an increased suicide risk compared to the general population in Germany [6].

For veterinary students, the experiences of depression and suicidal ideation have been less well examined. Due to the findings among veterinarians, it has recently become a focus of international scientific research [17–27], but for German veterinary students data was lacking so far. The results of this study provide data about depression, first data about suicidal ideation and suicide risk in German veterinary students compared to the German general population. These results clearly show that veterinary students have an increased risk of depression, suicidal ideation and suicide risk compared with the general population in Germany.

### Depression

The prevalence of depression was assessed with the PHQ-9, a very well-established self-report instrument. Our results regarding depression corresponds to the findings form other studies.

Of the German veterinary students 45.9% were screened positive for depression compared to only 3.2% for depression in the German general population. For US first-year veterinary students Hafen et al. conducted both a cross-sectional investigation [23] and a longitudinal investigation [25]: 32.0% of the first-semester students and 28.0% of the second-semester US veterinary students experienced clinical levels of depression. In addition to Hafen et al., studies that investigated US veterinary students in all years of study found comparable results. Kogan et al. [20] described that 37.3% of the surveyed US veterinary students in any given year report feeling depressed. Similar results were found by Karaffa and Hancock (2019). In their study, 33.9% of the surveyed US veterinary students reported moderate to severe levels of depression [27].

In 2017, Killinger et al. [17] found that 66.4% of veterinary students in North American suffer from depressive symptoms in all years of study. According to several studies from different countries, the rates of depression among veterinary students are higher than those in the general population [22, 23]. These findings correspond to the results of Schwerdtfeger et al. for German veterinarians. While only 3.9% of the German general population were screened positive for depression, the percentage of the German veterinarians was 27.8% [6]. Compared to German veterinarians, the rates of depression among German veterinary students are significantly higher (27.8% vs. 45.9%). As described by Killinger et al., the rates of depression vary by gender and age. Female students reported higher rates of depression than male students. Second- and third-year students showed higher depression rates than fourth-year students [17].

For veterinarians, long working hours including nightshifts and weekend services, professional and social isolation and personality traits are frequently mentioned as potential risk factors for depression [1, 5, 50]. According to Zenner et al. (2005) veterinary students have specific risk factors due to their studies, which are both personal, like a high willingness to perform, and structural, like admission requirements, high learning and examination load, associated social isolation and stress [24]. For US first-year students Hafen et al. (2006 and 2008) reported concerns about academic performance, financial concerns, the feeling of being behind in studies, time spent studying, and heavy workload as most distressing. Hafen et al. (2006 and 2008) note that homesickness, physical health, unclear professor expectations, academic concerns and difficulty fitting in with peers as strong predictors for depression [23, 25]. In Accordance with Kogan [20] struggling students begin to sacrifice their private lives to devote more time for studying. Stoewen (2019) summarizes that a chronic imbalance between job demands and one's resources to complete the job causes unresolvable long-term work-related stress. Furthermore, this can lead to burnout and compassion fatigue as a state of physical, emotional and mental exhaustion in combination with by doubts about the own competence and the value of one's work [51, 52]. Additionally, different studies resumes that chronicity of these feelings and behaviors during the whole training can lead to a decreased study-life balance and social isolation as well as stress, anxiety and depression [18–20]. For German veterinary students information on potential risk factors is lacking so far and further investigation is strongly recommended.

## Suicidal ideation and suicide risk

To assess suicidal ideation, Item 9 of the PHQ-9 was used. The degree of suicide risk was assessed using the SBQ-R. In the present study, 19.9% of the German veterinary students were classified as having current suicidal ideation and 24.0% showed an increased suicide risk compared to only 4.5% for suicidal ideation and 6.6% for suicide risk in the German general population. Our results reflect the findings of studies from several countries among veterinary students. In 2019 Karaffa and Hancock stated that over 30% of the surveyed US veterinary

students reported having seriously thought about attempting suicide and that almost 5% indicated having made a suicide attempt [27]. Skipper and Williams (2012) surveyed US veterinary students as well as veterinarians. The proportion of participants who claimed to have seriously considered or attempted suicide was 13.0% among veterinary students and 24.0% among licensed veterinarians [3]. For UK veterinary students Cardwell et al conducted a cross-sectional study in 2013. The results showed that 25.0% of the students reported suicidal ideation. According to Bartram et al. this percentage was comparable to 21.3% reported among UK veterinarians [9]. Both proportions were significantly higher compared to 16.7% in the English general population. 2.7% of the veterinary students stated having made a suicide attempt. This was significantly lower than the proportion of 5.6% of the English general population [26]. Cardwell et al. summarized that veterinary students and veterinarians were more likely to have suicidal thoughts, but less likely to have made an attempt, than members of the general population. However, the lower rate of suicide attempts among veterinarians might be due to the fact that veterinarians are more likely not to survive the attempt because of their knowledge performing euthanasia and the lethal drugs available to them, and therefore the rate of failed suicide attempts is lower among them than in the general population [53–55]. Platt et al. [56] reviewed preliminary studies, which indicated that veterinarians have a three times higher risk to die by suicide compared to the general population in several countries including the UK, the US, Belgium, Norway and Australia. For German veterinarians, the prevalence of suicidal ideation and suicide risk are nearly similar to those of the German veterinary students found in the present study. Schwerdtfeger et al. [6] indicated that 19.2% of the German veterinarians were classified as having suicidal ideation and 32.1% were screened positive for having an increased suicide risk. In contrast to the German general population both values were significantly higher (19.2% vs. 5.7% and 32.1% vs. 6.6%). In contrast to Schwerdtfeger et al. [6] in veterinary students suicidal ideation is higher in female (20.6% female vs. 13.1% male). However, female and male veterinary students show a comparable proportion of suicide risk (24.4% female vs. 21.3% male). These results are comparable to other findings in the literature, which also identified higher rates of suicidal ideation in females [57–59]. Although suicide attempts are more common in female, more males die by suicide [59, 60]. According to Clement et al., individuals possibly avoid or delay seeking professional help for mental health problems due to stigma [61]. Whereby men and health professionals, among others, are disproportionately deterred by stigma [61]. These gender differences should be considered when planning prevention programs [60].

Different aspects for the remarkably increased suicide risk among veterinary students are discussed by various authors. According to Brscic et al. [11], the high demands of the study program, along with the academic and transitional stress, can have a relevant impact on veterinary students' mental health. Halliwell and Hoskin [62] stated that the high academic admission requirements lead to student cohorts that are mostly highly intelligent and gifted. Admission criteria for veterinary medicine are also very high in Germany. The main criterion is the average grade of the university admission qualification. A high proportion of places are awarded to those ranked best according to high school grades. This results in an admission of the best high school graduates. Zenner et al. [24] indicate that due to the selection for admission according to "best grade", first-year students, whose excellent academic performance has been a unique selling point up to now, now find themselves in the midst of efficient and competitive fellow students. As a result, competitive behavior among students increases and cooperation among students decreases [24]. Results of studies with US medical students and veterinary students indicated that many students have the feeling from not fitting in with peers [25]. According to Joiner's interpersonal psychological theory of suicide (IPTS) [63] the feeling of not belonging to an appreciative group (thwarted belongingness) is highly relevant for

developing suicidal ideation. It is likely that reasons are similar among German veterinary students.

In conclusion, the prevalence of depression, suicidal ideation and suicide risk among veterinary students in Germany is significantly higher than in the German general population. Since the present study is a mere descriptive analysis of the data, assumptions about potential causes of increased prevalence cannot be made. We strongly recommend further investigation of possible risk factors and reasons for these increase rates to develop preventive measures and supporting programs for German veterinary students.

## Limitations

Although the population-based approach and the reference data of the German general population within our study are major strengths, several limitations to this study need to be considered. First, the proportion of female veterinary students in our sample is higher compared to the percentage of females in the entire population of German veterinary students. Thus, the proportion of male veterinary students in our sample is very low. Therefore, all findings on gender differences should be handled with caution. Second, women tend to report more mental distress than men [47–49] and the voluntary nature of participation carries the risk of bias due to an increased proportion of participants who may suffer from mental distress and may have personal experiences relating to the topic of this study. Third, suicidal ideation was assessed by self-report using Item 9 of the PHQ-9. Possibly it makes a difference whether suicidal ideation is assessed by self-report or in a face-to-face interview. However, since we are using a non-clinical sample and there is no therapeutic relationship, we decided to utilize the self-report. The methodology used is internationally established [64] and the comparative data are collected in a similar way. Fourth, when comparing the results of our study with those of the other studies cited, it must be considered that different methods and instruments were used and the studies were carried out in different years. Therefore, comparisons should be interpreted with caution.

## Supporting information

**S1 Table. Characteristics of participants.**
(DOCX)

**S1 Data.**
(SAV)

## Acknowledgments

The authors are thankful for the support by the five German Faculties for Veterinary Education and the veterinary medical student council initiatives in Germany in distributing the survey. The authors are very grateful for all veterinary students participating in this study.

## Author Contributions

**Conceptualization:** Nadine Schunter, Heide Glaesmer, Mahtab Bahramsoltani.

**Data curation:** Heide Glaesmer.

**Formal analysis:** Luise Lucht.

**Investigation:** Nadine Schunter.

**Methodology:** Nadine Schunter, Heide Glaesmer, Luise Lucht, Mahtab Bahramsoltani.

**Project administration:** Heide Glaesmer, Mahtab Bahramsoltani.

**Supervision:** Heide Glaesmer, Mahtab Bahramsoltani.

**Writing – original draft:** Nadine Schunter.

**Writing – review & editing:** Heide Glaesmer, Luise Lucht, Mahtab Bahramsoltani.

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
