## [Decision Letter · Decision Letter 0]

19 Mar 2022

PONE-D-21-39300Submission of a manuscript entitled “Depression, suicidal ideation and suicide risk in German veterinary medical students compared to the German general population”PLOS ONE

Dear Dr. Schunter,

Thank you for submitting your manuscript to PLOS ONE. After careful consideration, we feel that it has merit but does not fully meet PLOS ONE’s publication criteria as it currently stands. Therefore, we invite you to submit a revised version of the manuscript that addresses the points raised during the review process.

Reviewers have highlighted that your paper is an interesting work and I agree with them. However, reviewers have pointed out some issues that should be resolved prior to publication acceptance. Please, consider their comments.

We look forward to receiving your revised manuscript.

Kind regards,

José J. López-Goñi

Academic Editor

PLOS ONE

Journal Requirements:

3. Please provide additional details regarding participant consent. In the ethics statement in the Methods and online submission information, please ensure that you have specified what type you obtained (for instance, written or verbal, and if verbal, how it was documented and witnessed). If your study included minors, state whether you obtained consent from parents or guardians. If the need for consent was waived by the ethics committee, please include this information

4. During your revisions, please note that a simple title correction is required: as written in the submission form, the current title erroneously includes the phrase "Submission of a manuscript entitled..." Please ensure this is updated in the manuscript file and the online submission information.

6. We note that you have referenced (ie. Bewick et al. [5]) which has currently not yet been accepted for publication. Please remove this from your References and amend this to state in the body of your manuscript: (ie “Bewick et al. [Unpublished]”) as detailed online in our guide for authors

Additional Editor Comments (if provided):

Reviewers have highlighted that your paper is an interesting work and I agree with them. However, reviewers have pointed out some issues that should be resolved prior to publication acceptance. Please, consider their comments.

Reviewers' comments:

Reviewer's Responses to Questions

**Comments to the Author**

1. Is the manuscript technically sound, and do the data support the conclusions?

Reviewer #1: Yes

Reviewer #2: Partly

Reviewer #3: Yes

2. Has the statistical analysis been performed appropriately and rigorously? 

Reviewer #1: Yes

Reviewer #2: Yes

Reviewer #3: Yes

3. Have the authors made all data underlying the findings in their manuscript fully available?

Reviewer #1: Yes

Reviewer #2: Yes

Reviewer #3: Yes

4. Is the manuscript presented in an intelligible fashion and written in standard English?

Reviewer #1: Yes

Reviewer #2: Yes

Reviewer #3: Yes

5. Review Comments to the Author

Reviewer #1: The present study aims to evaluate the presence of depression, suicidal ideation and suicide risk in veterinary medical students in Germany, since there is evidence that veterinarians have a significantly higher risk of suffering from these problems. The authors concluded that German veterinary medical students have a higher risk to be screened positive for depression, to report current suicidal ideation and to have an increased suicide risk compared with the general population in Germany of the same age range (18-46 years).

This kind of research is very useful and necessary. Therefore, the proposed work can be very interesting. I wish to compliment the authors on their thoughtful work and worthwhile goal.

Overall, the article is well written, and the logic of the study is according to the goal. The paper provides useful data, and the main findings are consistent. In addition, it is a novel study since it focuses on a specific population, making the particular situation of German veterinary medical students visible. Even so, some considerations and suggestions are provided below.

MAJOR CONCERNS

Method

For the evaluation of suicidal ideation, a single item of the PHQ-9 questionnaire is used. Authors are suggested to be careful in assessing this problem. Making use of a self-report can be dangerous and falsify the data due to the component of subjectivity. In addition, using an online questionnaire limits the information obtained. For future research, it could be solved with the use of an interview to collect information on suicidal ideation and risk.

MINOR CONCERNS

Introduction

A good review of the topic is presented and well structured.

Results

Authors are suggested to indicate percentages and n in the same way (page 14, paragraph 2).

It is not necessary to mention table 1 at the end of each paragraph of the results. Mentioning table 1 in the first paragraph of the results is sufficient.

Discussion

The discussion section is well presented. A good comparison is made with studies carried out in other countries, highlighting the novelty of the current study.

With these changes, readers will be able to fully appreciate the potential clinical significance of the findings and future directions for research. I hope these proposed modifications will serve to improve the manuscript.

Reviewer #2: State gen pop source more clearly in abstract

122 define council initiative

Make more clear the purpose of the gen pop sample. It does not seem to be for the purpose of the comparison in this paper, but is a dataset these authors accessed after the fact. <ore about="" and="" between="" collection="" data="" for="" gen="" in="" information="" initial="" is="" needed.="" of="" owners="" paper="" pop="" purpose="" relationship="" researchers="" set="" the="" this="">

Find a way to merge paragraphs about german gen pop sample 1 and 2 since much of the methodology was the same

194 “the online questionnaire used with the veterinary student population for this research…..”

218 remove economically… it is confusing… to reliably and validly assess would serve more merit on the reasoning for using this tool more over, the tool was used on the gen pop which sample so I imagine is the reason this tool was used with students?

224 remove “patients” as this tool is not being used in a mental health clinical population with either the DVM or gen pop samples for this research. If this tool is only used for mental health patients there should be some discussion of the validity of the tool for use with gen pop and veterinary populations.

244 If there is a statistical difference between the sample and the population of veterinary students regarding proportion of females, state it. The research is still valuable even with this caveat. The way this reads does not provide a clear description of the limitations of your sample. You can cite research which suggests females tend to report distress more readily than males as a possible reason for this discrepancy between your sample and the larger dvm sample…

357 when stating that attempts are lower in veterinary population it is important to acknowledge that may be due to the fact that the survey is collecting information from living people and that DVM’s are more successful at completing suicide than the gen pop. Attempts look as if they are less simply because of the sampling of living individuals instead of death records… see Tomasi, S. E., Fechter-Leggett, E. D., Edwards, N. T., Reddish, A. D., Crosby, A. E., & Nett, R. J. (2019). Suicide among veterinarians in the United States from 1979 through 2015. Journal of the American Veterinary Medical Association, 254(1), 104–112.

413 Limitations… consider discussing more fully in this and other sections… this topic Koopmans, G. T., & Lamers, L. M. (2007). Gender and health care utilization: the role of mental distress and help-seeking propensity. Social Science & Medicine, 64(6), 1216–1230.

Moreover, just because males may under report they more successfully complete suicide, so efforts to decrease stigma for help seeking is key for men… could be added to recommendations section…

Good job team!</ore>

Reviewer #3: 58 Keywords

I suggest: Occupational risk factors, psychological wellbeing

70: This and other potential risk factors were described by a recent paper where a wide set of extracted literature was submitted to text mining and topic modelling

76: veterinary students= veterinary medicine students; here and elsewhere I suggest you talk about veterinary students to shorten the sentence specifying the first time that for veterinary students you mean those enrolled in the veterinary program and not nurses and similar courses

110: Mention the large amount of studies on students in literature about here

145: “One participant indicated diverse gender”

? How did you control that? That should not be included

339: Please include a statement and consider compassion fatigue among the following

392: See paper by Brscic, M., Contiero, B., Schianchi, A., & Marogna, C. (2021). Challenging suicide, burnout, and depression among veterinary practitioners and students: text mining and topics modelling analysis of the scientific literature. BMC veterinary research, 17(1), 1-10.

423: Consider the different years and methods/instruments as well

6. PLOS authors have the option to publish the peer review history of their article (what does this mean?). If published, this will include your full peer review and any attached files.

Reviewer #1: No

Reviewer #2: No

Reviewer #3: No

---

## [Author Response · Author response to Decision Letter 0]

20 May 2022

Reviewer’s comments: 

Reviewer #1: 

Comment:

The present study aims to evaluate the presence of depression, suicidal ideation and suicide risk in veterinary medical students in Germany, since there is evidence that veterinarians have a significantly higher risk of suffering from these problems. The authors concluded that German veterinary medical students have a higher risk to be screened positive for depression, to report current suicidal ideation and to have an increased suicide risk compared with the general population in Germany of the same age range (18-46 years).

This kind of research is very useful and necessary. Therefore, the proposed work can be very interesting. I wish to compliment the authors on their thoughtful work and worthwhile goal.

Overall, the article is well written, and the logic of the study is according to the goal. The paper provides useful data, and the main findings are consistent. In addition, it is a novel study since it focuses on a specific population, making the particular situation of German veterinary medical students visible. Even so, some considerations and suggestions are provided below.

Answer: 

We would like to thank you for your positive perception of this article and our study. We really appreciate your feedback and your suggestions for correction.

MAJOR CONCERNS

Method

Comment:

For the evaluation of suicidal ideation, a single item of the PHQ-9 questionnaire is used. Authors are suggested to be careful in assessing this problem. Making use of a self-report can be dangerous and falsify the data due to the component of subjectivity. In addition, using an online questionnaire limits the information obtained. For future research, it could be solved with the use of an interview to collect information on suicidal ideation and risk.

Answer: 

Thank you very much for this comment. I'm sure this thought also crosses the minds of many readers of this article. We therefore adapted the method section, were we defined suicidal ideation as “being bothered by thoughts, that oneself would be better off dead, or of hurting oneself in some way, over the past two weeks” (Line 217-218; The information regarding the lines relates to the track change mode "Simple Markup"). 

Additionally, we included the aspect of self-report as a limitation in the limitation section (Line 439-444).

The assessment of suicidal thoughts is fundamentally a challenge. They are not accessible by objective measurement, and one therefore inevitably has to rely on self-report. The methodology used is internationally established and the comparative data are collected in a similar way. We believe that this is the best approach for this type of study, but we now discuss the limitations in more detail (Line 439-444). 

MINOR CONCERNS

Introduction

Comment:

A good review of the topic is presented and well structured.

Answer: 

Thank you very much for this comment.

Results

Comment:

Authors are suggested to indicate percentages and n in the same way (page 14, paragraph 2).

Answer: 

Thank you for the remark. We have adjusted the indication of percentages and n. (Line 285, page 15, paragraph 2).

Comment:

It is not necessary to mention table 1 at the end of each paragraph of the results. Mentioning table 1 in the first paragraph of the results is sufficient.

Answer: 

Thank you for this advice. We deleted the redundant references to Table 1 (Line 280, Line 294 and Line 303).

Discussion

Comment:

The discussion section is well presented. A good comparison is made with studies carried out in other countries, highlighting the novelty of the current study.

With these changes, readers will be able to fully appreciate the potential clinical significance of the findings and future directions for research. I hope these proposed modifications will serve to improve the manuscript.

 Answer: 

Thank you very much. We are sure that your comments will improve the manuscript.

Reviewer #2: 

Comment:

State gen pop source more clearly in abstract

Answer: 

Thank you for this remark. In Abstract we added the source of the of the German general population samples by describing that they were collected with the assistance of a demographic consulting company (Line 39-40; The information regarding the lines relates to the track change mode "Simple Markup").

Comment:

122 define council initiative

Answer: 

Thank you for pointing this out. We added the description of the student council initiative to the paper (Line 123-125).

Comment:

Make more clear the purpose of the gen pop sample. It does not seem to be for the purpose of the comparison in this paper, but is a dataset these authors accessed after the fact. Find a way to merge paragraphs about german gen pop sample 1 and 2 since much of the methodology was the same 

Answer: 

Thank you very much for this comment. We described of the purpose of using the representative general population samples in more detail (Line 158-161). In addition, we merged the two paragraphs about the German general population samples (Line 163-192). 

Comment:

194 “the online questionnaire used with the veterinary student population for this research…..”

Answer: 

Thank you for the concretisation. As recommended, we adjusted the text accordingly (Line 195). 

Comment:

218 remove economically… it is confusing… to reliably and validly assess would serve more merit on the reasoning for using this tool more over, the tool was used on the gen pop which sample so I imagine is the reason this tool was used with students?

Answer: 

Yes, the assumption is correct. The SBQ-R is a reliable and valid instrument for assessing different aspects of suicidality. Therefore, it was used in the population sample surveys as well as in our study. As recommended, we replaced " economically " with "reliable and valid" (Line 224).

Comment:

224 remove “patients” as this tool is not being used in a mental health clinical population with either the DVM or gen pop samples for this research. If this tool is only used for mental health patients there should be some discussion of the validity of the tool for use with gen pop and veterinary populations. 

Answer: 

Thank you for the remark. We agree with you and replaced “patients” by “participants” (Line 233). The major advantage of the SBQ-R is that it can be used not only with psychiatric inpatient, but also with nonclinical participants. Its use is also recommended for population-based studies. For a more explicit explanation, we have included these two aspects (Lines 225-227).

Comment:

244 If there is a statistical difference between the sample and the population of veterinary students regarding proportion of females, state it. The research is still valuable even with this caveat. The way this reads does not provide a clear description of the limitations of your sample. You can cite research which suggests females tend to report distress more readily than males as a possible reason for this discrepancy between your sample and the larger dvm sample… 

Answer: 

Thank you for the comment. As recommended, we have addressed the difference in the proportion of females between our sample and the overall population of veterinary students. We followed your comment and explained this aspect by the possibility that females report mental distress more often than males citing Koopmans (2007), Hatch (2011) and Jatrana, (2021) (Lines 249-254).

Comment:

357 when stating that attempts are lower in veterinary population it is important to acknowledge that may be due to the fact that the survey is collecting information from living people and that DVM’s are more successful at completing suicide than the gen pop. Attempts look as if they are less simply because of the sampling of living individuals instead of death records… see Tomasi, S. E., Fechter-Leggett, E. D., Edwards, N. T., Reddish, A. D., Crosby, A. E., & Nett, R. J. (2019). Suicide among veterinarians in the United States from 1979 through 2015. Journal of the American Veterinary Medical Association, 254(1), 104–112.

Answer: 

Thank you very much for this comment. We followed your recommendation and specified the aspect of “successful” suicide attempts, citing Platt (2012), Nett (2015) and Tomasi (2019) (Lines 384-387).

Comments:

413 Limitations… consider discussing more fully in this and other sections… this topic

Koopmans, G. T., & Lamers, L. M. (2007). Gender and health care utilization: the role of mental distress and help-seeking propensity. Social Science & Medicine, 64(6), 1216– 1230.

Answer: 

Answer: 

Thank you for the remark. For a more stringent argumentation, we added the difference in the proportion of females between our sample and the overall population of veterinary students as well as the aspect that women tend to report mental distress more often (Line 433-437).

Comment:

Moreover, just because males may under report they more successfully complete suicide, so efforts to decrease stigma for help seeking is key for men… could be added to recommendations section…

Answer: 

Thank you for pointing this out. We added the aspects that death by suicide is more common in males and that stigma may be a deterrent to help seeking. Whereby men are more deterred by stigma (Line 400-404). 

Comment:

Good job team!

 

Reviewer #3: 

Comment: 

58 Keywords, I suggest: Occupational risk factors, psychological wellbeing

Answer: 

Thank you for the amendment. We included the suggested keywords. (Line 53-54; The information regarding the lines relates to the track change mode "Simple Markup").

Comment

70: This and other potential risk factors were described by a recent paper where a wide set of extracted literature was submitted to text mining and topic modelling, See paper by Brscic, M., Contiero, B., Schianchi, A., & Marogna, C. (2021). Challenging suicide, burnout, and depression among veterinary practitioners and students: text mining and topics modelling analysis of the scientific literature. BMC veterinary research, 17(1), 1-10.

Answer: 

Thank you for recommending this valuable reference. We added further potential risk factors from the paper by Brscic et al. in the manuscript (Line 66-68) and pointed out, that they are specific for veterinary practitioners and could not be readily applied to vet students (Line 90-93).

Comment:

76: veterinary students= veterinary medicine students; here and elsewhere I suggest you talk about veterinary students to shorten the sentence specifying the first time that for veterinary students you mean those enrolled in the veterinary program and not nurses and similar courses

Answer:

Thank you for this remark. We replaced “veterinary medical students” by “veterinary students” in the manuscript and specified, that they are enrolled in the veterinary program (Line 72-73).

Comment:

110: Mention the large amount of studies on students in literature about here 

Answer: 

Thank you for this comment. We added the information of the large amount of studies from several countries (Line 111).

Comment:

145: “One participant indicated diverse gender” ? How did you control that? That should not be included

Answer:

Thank you for your thoughtful advice. As we were conscious of not being able to include the participant with diverse gender, we excluded this participant for all gender group analyses and only included the respective data in the descriptive statistics of the total population.

Comment:

339: Please include a statement and consider compassion fatigue among the following 

Answer:

Thank you for this addition. We included the aspect of developing compassion fatigue (Line 355-359).

Comment:

392: See paper by Brscic, M., Contiero, B., Schianchi, A., & Marogna, C. (2021). Challenging suicide, burnout, and depression among veterinary practitioners and students: text mining and topics modelling analysis of the scientific literature. BMC veterinary research, 17(1), 1-10.

Answer:

Thank you for this comment. We added further possible stressors stated by Brscic et al. (Line 406-408).

Comment:

423: Consider the different years and methods/instruments as well

Answer:

Thank you very much for this comment. Assessing suicidal thoughts is fundamentally challenging, because it cannot be measured objectively. Therefore, one must inevitably rely on self-reporting. We assessed suicidal ideation by self-report using Item 9 of the PHQ-9, as it is an internationally established methodology. We now discuss the aspect of self-report as a limitation in more detail in the limitations section (Line 439-444). In addition, we added the aspect of the different methods and instruments used in our study and the studies we cited (Line 444-447).

---

## [Decision Letter · Decision Letter 1]

20 Jun 2022

Depression, suicidal ideation and suicide risk in German veterinary medical students compared to the German general population

PONE-D-21-39300R1

Dear Dr. Schunter,

We’re pleased to inform you that your manuscript has been judged scientifically suitable for publication and will be formally accepted for publication once it meets all outstanding technical requirements.

Kind regards,

José J. López-Goñi

Academic Editor

PLOS ONE

Additional Editor Comments (optional):

Reviewers' comments:

Reviewer's Responses to Questions

**Comments to the Author**

1. If the authors have adequately addressed your comments raised in a previous round of review and you feel that this manuscript is now acceptable for publication, you may indicate that here to bypass the “Comments to the Author” section, enter your conflict of interest statement in the “Confidential to Editor” section, and submit your "Accept" recommendation.

Reviewer #1: All comments have been addressed

2. Is the manuscript technically sound, and do the data support the conclusions?

Reviewer #1: Yes

3. Has the statistical analysis been performed appropriately and rigorously? 

Reviewer #1: Yes

4. Have the authors made all data underlying the findings in their manuscript fully available?

Reviewer #1: Yes

5. Is the manuscript presented in an intelligible fashion and written in standard English?

Reviewer #1: Yes

6. Review Comments to the Author

Reviewer #1: After a second revision, the authors have fully implemented the proposed changes.

I congratulate the authors for their work and trust that the recommendations and proposed changes have been helpful.

7. PLOS authors have the option to publish the peer review history of their article (what does this mean?). If published, this will include your full peer review and any attached files.

Reviewer #1: No

---

## [Editor Report · Acceptance letter]

4 Aug 2022

PONE-D-21-39300R1 

Depression, suicidal ideation and suicide risk in German veterinary medical students compared to the German general population 

Dear Dr. Schunter:

I'm pleased to inform you that your manuscript has been deemed suitable for publication in PLOS ONE. Congratulations! Your manuscript is now with our production department. 

Kind regards, 

on behalf of

Dr. José J. López-Goñi 

Academic Editor

PLOS ONE